# LEARNING REPRESENTATIONS FOR BINARY-CLASSIFICATION WITHOUT BACKPROPAGATION

**Mathias Lechner**
IST Austria
Am Campus 1, Klosterneuburg, Austria
mlechner@ist.ac.at

## ABSTRACT

The family of feedback alignment (FA) algorithms aims to provide a more biologically motivated alternative to backpropagation (BP), by substituting the computations that are unrealistic to be implemented in physical brains. While FA algorithms have been shown to work well in practice, there is a lack of rigorous theory proofing their learning capabilities. Here we introduce the first feedback alignment algorithm with provable learning guarantees. In contrast to existing work, we do not require any assumption about the size or depth of the network except that it has a single output neuron, i.e., such as for binary classification tasks. We show that our FA algorithm can deliver its theoretical promises in practice, surpassing the learning performance of existing FA methods and matching backpropagation in binary classification tasks. Finally, we demonstrate the limits of our FA variant when the number of output neurons grows beyond a certain quantity.

## 1 INTRODUCTION

A key factor enabling the successes of Deep Learning is the backpropagation of error (BP) algorithm (Rumelhart et al., 1986). Since it has been introduced, BP has sparked several discussions on whether physical brains are realizing BP-like learning or not (Grossberg, 1987; Crick, 1989). Today, most researchers consent that two distinct characteristics of BP render the idea of a BP based learning in brains as implausible: 1) The usage of symmetric forward and backward connections and 2) the strict separation of activity and error propagation (Bartunov et al., 2018). These two objections have lead researchers to search for more biologically motivated alternatives to BP.

The three most influential families of BP alternatives distilled so far are Contrastive Hebbian Learing (CHL) (Movellan, 1991), target-propagation (TP) (LeCun, 1986; Hinton, 2007; Bengio, 2014) and feedback Alignment (FA) (Lillicrap et al., 2016).

The idea of CHL is to propagate the target activities, instead of the errors, backward through the network. For this reason, a temporal dimension is added to the neuron activities. Each neuron then adapts its parameters based on the temporal differences of its "forward" and "backward" activity. The two significant critic points of CHL are the requirement for symmetric "forward-backward" connections and the use of alternating "forward" and "backward" phases (Baldi & Pineda, 1991; Bartunov et al., 2018).

TP shares the idea with Contrastive Hebbian Learning of propagating target activities instead of errors. However, rather than keeping symmetric forward and backward paths, the reciprocal propagation of the activities are realized through learned connections. Consequently, each layer has assigned two objectives: Learning the inverse of the layer's forward function and minimizing the difference to the back-projected target activity. Variants of TP differ in how exactly the target activity is projected backward (LeCun, 1986; Bengio, 2014; Bartunov et al., 2018). Theoretical guarantees of TP rely on the assumption that each reciprocal connection implements the perfect inverse of the corresponding forward function. This issue of an imperfect inverse was also found to be the "bottleneck" of TP in practice (Bartunov et al., 2018). When the output of a layer has a significant lower dimension than its input, reconstructing the input from the output becomes challenging, resulting in poor learning performance.

Feedback alignment algorithms eliminate the weight sharing implausibility of BP by replacing the symmetric weights in the error backpropagation path by random matrices. The second objection, i.e., separate activity and error channels, is attenuated by Direct Feedback Alignment (Nøkland, 2016) which drastically reduces the number of channels carrying an error signal. While feedback alignment algorithms work well on small and medium-sized benchmarks, a recent study identified that they are unable to provide learning on more challenging datasets like ImageNet (Bartunov et al., 2018). Another criticism of FA algorithms is the lack of rigorous mathematical justification and convergence guarantees of the performed computations.

In this work, we investigate feed-forward networks where the weights of all, expect the first, layers are constrained to positive values. We prove that this constraint does not invalidate the universal approximation capabilities of neural networks. Next, we show that, in combination with monotonic activation functions, all layers from the second layer on realize monotonically increasing functions. The backpropagation of a scalar error signal through these layers only affects the magnitude of the error signal but does not change its sign. Consequently, we prove that algorithms that bypass the error backpropagation steps, such as Direct Feedback Alignment, can compute the sign of the true gradient with respect to the weights of our constraint networks without the need for backpropagation. Finally, we show that our algorithm, which we call monotone Direct Feedback Alignment, can deliver its theoretical promises in practice, by surpassing the learning performance of existing feedback alignment algorithms in binary classification task, i.e., when the error signal is scalar, and provide decent performance even when the error signal is not scalar.

We make the following key contributions:

- First FA algorithm that has provable learning capabilities for non-linear networks of arbitrary depth

- Experimental evaluation showing that our FA algorithm outperforms the learning performance of existing FA algorithms and match backpropagation in binary classification tasks

- We make an efficient TensorFlow implementation of all tested algorithms publicly available[1]

## 2 BACKPROPAGATION AND FEEDBACK ALIGNMENT

We consider the feed-forward neural network

$$
h_l(h_{l-1}) := \begin{cases} f(W_l h_{l-1} + b_l) & \text{if } l < L \\ W_l h_{l-1} + b_l & \text{if } l = L \end{cases}
$$
$$
h_0 := x \tag{1}
$$
$$
W_l \in \mathbb{R}^{n_l \times n_{l-1}},
$$
$$
b_l \in \mathbb{R}^{n_l}
$$

where $f$ is the non-linear activation function, $x$ the input and $h_L$ the output of the network. For classification tasks, $h_L$ is usually transformed into a probability distribution with discrete support by a sigmoid or softmax function.

During training, the parameters $W_l, b_l, l = 1, \ldots L$ are adjusted to minimize a loss function $\mathcal{L}(y, h_L)$ on samples of a giving training distribution $p(y, x)$. This is usually done by performing *gradient descent*

$$
\theta_l \leftarrow \theta_l - \alpha \frac{d\mathcal{L}}{d\theta}, \qquad \alpha \in \mathbb{R}_+ \tag{2}
$$

with respect to the parameters $\theta_l \in \{W_l, b_l\}, 1 \leq l \leq n$ of the network.

---

[1]https://github.com/mlech26l/iclr_paper_mdfa

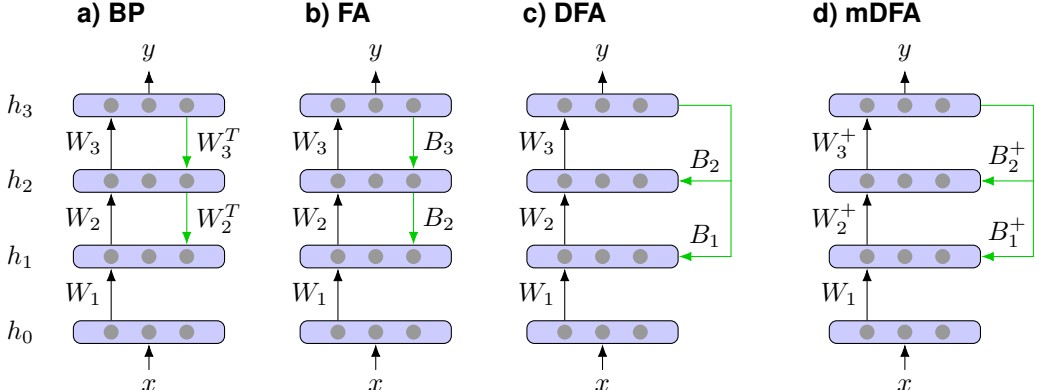

Figure 1: Graphical overview of various error transport methods, adapted from Nøkland (2016). Black arrows represent the forward path, whereas green arrows represent the error feedback. Weights that are learned are denoted as $W_i$ and fixed weights are denoted by $B_i$. Weights that are restricted to only positive values are annotated by a $^+$. **a)** Backpropagation, **b)** Feedback alignment, **c)** Direct Feedback Alignment, **d)** monotone Direct Feedback Alignment.

## 2.1 BACKPROPAGATION

Backpropagation (Rumelhart et al., 1986) is the primary method to compute the gradients needed by the updates in equation (2) by iteratively applying the chain-rule

$$\frac{d\mathcal{L}}{d\theta_l} = \left(\frac{dh_l}{d\theta_l}\right)^T \frac{d\mathcal{L}}{dh_l} \tag{3}$$

$$\frac{d\mathcal{L}}{dh_l} = \left(\frac{dh_{l+1}}{dh_l}\right)^T \frac{d\mathcal{L}}{dh_{l+1}} \tag{4}$$

$$\frac{dh_{l+1}}{dh_l} = W_{l+1}\text{diag}(f'(W_{l+1}h_l + b_{l+1})). \tag{5}$$

A graphical representation of how information first flow forward and then backward in BP through each layer is shown in Figure (1) a.

Two major concerns argue against the idea that biological neural networks are implementing BP-based learning. I) The weight matrix $W_l$ of the forward path is reused in the backward path in the form of $W_l^T$ (*weight sharing*), and II) the strict separation of activity carrying forward and error carrying backward connections (*reciprocal error transport*).

## 2.2 FEEDBACK ALIGNMENT ALGORITHMS

Feedback alignment addresses the implausibility of reusing $W_l^T$ in the backward path by replacing $W_l^T$ by a fixed random matrix $B_l$. Lillicrap et al. (2016) showed that this somewhat counterintuitive approach works remarkably well in practice. The term "feedback alignment" originates from the observations made by Lillicrap et al. (2016) that the angle between the FA update vector and the true gradient starts to decrease, i.e., *align*, after a few epochs of the training algorithm. Theoretical groundwork on this alignment principle of FA relies on strong assumptions such as a linearized network with one hidden layer (Lillicrap et al., 2016).

FA avoids any *weight sharing* but does not address the *reciprocal error transport* implausibility, due to its strict separation of forward and backward pathways, as shown in Figure (1) b. Direct Feedback Alignment (DFA) (Nøkland, 2016) relaxes this issue by replacing all backward paths with a direct feed from the output layer error-gradient $\frac{d\mathcal{L}}{dh_L}$. Consequently, there is only a single error signal that is distributed across the entire network, which is arguably more biologically plausible than reciprocal error connections. The resulting parameter updates of DFA are of the form

$$\delta\theta_l := \begin{cases} \frac{d\mathcal{L}}{dh_L}\frac{h_L}{\theta_l} & \text{if } l = L \\ \frac{d\mathcal{L}}{dh_L}B_l\frac{h_l}{\theta_l} & \text{if } l < L \end{cases} \tag{6}$$

, where $B_l \in \mathbb{R}^{n_L \times n_l}$ is a random matrix. A graphical schematic of DFA is shown in Figure (1) c. Similar to FA, DFA shows a decent learning performance in mid sized classification tasks (Nøkland, 2016), but fails on more complex datasets such as ImageNet (Bartunov et al., 2018). Theory on adapting the alignment principle to DFA shows that under the strong assumptions of constant DFA update directions and a layer-wise criterion minimization, the DFA update vector will align with the true gradient (Nøkland, 2016; Gilmer et al., 2017).

Recently, Frenkel et al. (2019) proposed to combine ideas from feedback alignment and target-propagation in their Direct Random Target Projection (DRTP) algorithm. While DRTP shows decent empirical performance, theoretical guarantees about DRTP rely on linearized networks.

## 2.3 SIGN-SYMMETRY ALGORITHMS

Liao et al. (2016) introduced the sign-symmetry algorithm, a hybrid of BP and FA. Sign-symmetry locks the signs of the feedback weight $B_l$ to have the same signs as $W_l^T$, but random absolute value. The authors showed that this approach drastically improves learning performance compared to standard FA. Furthermore, Moskovitz et al. (2018) and Xiao et al. (2019) demonstrated that the sign-symmetry algorithm is even able to match backpropagation for training deep network architectures and large datasets such as ImageNet.

While these empirical observations suggest that the polarity of the error feedback is more important than its magnitude, the mathematical justification of sign-symmetry remains absent. Similar to FA, sign-symmetry relaxes the strict *weight sharing* implausibility, but still relies on an unrealistic reciprocal error transport.

## 3 MONOTONE DIRECT FEEDBACK ALIGNMENT

In this section, we first introduce a new class of feed-forward networks, where all, except the first, layers are constrained to realize monotone functions. We call such networks *mono-nets* and show that they are as expressive as unconstrained feed-forward networks. Next, we prove that for our mono-nets with single output tasks, e.g., binary-classification, feedback alignment algorithms provide the sign of the gradient. The sign of the gradient is interesting for learning, as it tells us if the value of a parameter should be increased or decreased in order to reduce the loss. At the end of this section, we will highlight similarities to algorithms from literature, which can provide resilient learning by only relying on the sign of the gradient.

Neural networks with monotonic constraints have been already studied in literature (You et al., 2017), however not in the context of learning algorithms.

**Definition 1** (mono-net). *A mono-net is a feed-forward neural network with $L$ layers $h_1, \ldots h_L$, each layer $l$ composted of $n_l$ units and the semantics*

$$h_l(h_{l-1}) := \begin{cases} f(W_l h_{l-1} + b_l) & \text{if } l < L \\ W_l h_{l-1} + b_l & \text{if } l = L \end{cases} \tag{7}$$

$$h_0 := x \tag{8}$$

$$W_1 \in \mathbb{R}^{n_1 \times n_0}, \tag{9}$$

$$W_l \in \mathbb{R}_+^{n_l \times n_{l-1}}, \text{ for } l > 1 \tag{10}$$

$$b_l \in \mathbb{R}^{n_l} \tag{11}$$

*where $\mathbb{R}_+$ are the positive reals, i.e. $\mathbb{R}_+ = \{x \in \mathbb{R} | x \geq 0\}$, $f$ is a non-linear monotonic increasing activation function, $x$ the input and $h_L$ the output of the network.*

The major difference between mono-nets and general feed-forward neural networks is the restriction to only positive weight values in layers from the second layer on. Combined with the monotonic increasing activation function, this means that each layer $h_l(h_{l-1}), l \geq 2$ realizes a monotone increasing function. Because functional composition preserves monotonicity, the complete network up to the first layer

$$h_l \circ h_{l-1} \circ \cdots \circ h_2(h_1) \tag{12}$$

implements a monotone increasing function.

**mono-nets are Universal Approximators** At first glance, this restriction seems counterproductive, as it might interfere with the expressiveness of the networks. However, we proof in Theorem 1 that our mono-nets with tangent hyperbolic activation are universal approximators, meaning that we can approximate any continuous function arbitrarily close. A potential drawback of the monotonicity constraint is that we might need a larger number of units in the hidden layers to achieve the same expressiveness as a general feed-forward network, as illustrated in our proof of Theorem 1.

**Theorem 1** (mono-nets are Universal Approximators). *Let $I_n$ be the $n$-dimensional unit hypercube $[0,1]^n$ and $C(I_n)$ denote the set of continuous functions $f : I_n \to \mathbb{R}$. We define $\|f\|_\infty$ as the supremum norm of $f \in C(I_n)$ over its domain $I_n$. For any given $f \in C(I_n)$ and $\varepsilon > 0$, there exist a function $m : I_n \to \mathbb{R}$ of the form*

$$m(x) := \sum_{i=1}^{M} \bar{v}_i \tanh(\bar{w}_i^T x + \hat{w}_i^T(-x) + \bar{b}_i) + c \qquad (13)$$

*with $\bar{v} \in \mathbb{R}_+{}^M, \bar{w}_i \in \mathbb{R}_+{}^n, \hat{w}_i \in \mathbb{R}_+{}^n, \bar{b} \in \mathbb{R}^M, c \in \mathbb{R}$ and $M < \infty$ such that*

$$\|m(x) - f(x)\|_\infty < \varepsilon.$$

*In essence, the set of functions $m(x)$ of the form given in (19) is* dense *in $C(I_n)$.*

*Proof.* See supplementary materials 3. $\qquad\qquad\square$

**mDFA provides the sign of the gradient** Here, we prove that for 1-dimensional outputs DFA applied on a mono-net, which we will call simply mDFA, provides the sign of the true gradient. Note that we focus our methods on DFA instead of "vanilla" FA, due to the superiority of DFA in terms of biological plausibility and empirical performance (Nøkland, 2016; Bartunov et al., 2018).

**Theorem 2** (For 1-dimensional outputs mDFA computes the sign of the gradient). *Let $\mathcal{L}(y, h_L)$ be a loss function, $m(x) := h_L \circ h_{L-1} \circ \cdots \circ h_2 \circ h_1 \circ h_0(x)$ be a mono-net according to Definition 1 with parameters $\Theta := \{W_l, b_l | l = 1, \ldots L\}$. We denote $\delta\theta$ the update value computed by mDFA and $\nabla\theta$ as the gradient $\frac{\partial \mathcal{L}}{\partial \theta}$ for any $\theta \in \{W_l, b_l\}$ with $1 \le l \le L$. If $n_L = 1$, it follows that*

$$\left(\delta\theta\right)_{i,j} \cdot \left(\nabla\theta\right)_{i,j} \ge 0, \qquad (14)$$

*for each coordinate $(i, j)$ of $\theta$.*

*Proof.* See supplementary materials. $\qquad\qquad\square$

A graphical illustration of how activities and errors propagate in mDFA is shown in Figure (1) d.

**Literature on learning by relying only on the sign of the gradient** Two learning concepts related to mDFA are RPROP (Riedmiller & Braun, 1993; Riedmiller, 1994) and signSGD (Bernstein et al., 2018). RPROP aims to build a more resilient alternative to gradient descent by decoupling the amplitude of the gradient from the step size of parameter updates. In essence, for each coordinate RPROP adapts the step size based on the sign of the most recent gradients computed. Riedmiller & Braun (1993) showed that their approach could stabilize the training of a neural network compared to standard gradient descent.

Performing gradient descent with taking the sign of each gradient coordinate is on an algorithmic level equivalent to the well-known *steepest descent method* with $L_\infty$ norm (Boyd & Vandenberghe, 2004; Bernstein et al., 2018). signSGD (Bernstein et al., 2018) studies convergence properties of the stochastic approximation of this algorithm.

**What about networks with more than one output neuron?** Theorem 2 applies only to networks with scalar output. As a natural consequence, one may ask whether such theoretical guarantees can be extended to more dimensional output variables. The simple answer is, unfortunately not. In the supplementary materials section A.3 we provide a counterexample showing that Theorem 2 naively extended to two output neurons does not hold anymore.

We want to note that the requirement of a neural network to have only a single output neuron is biologically unjustified. It is known that sub-circuits of biological neuronal networks can feed to multiple motor neuron groups Cook et al. (2019).

**How does mDFA relate to the non-negative matrix factorization?** A seemingly related concept to mDFA is the non-negative matrix factorization (NMF) algorithm. NMF decomposes an observation matrix $V$ into a weight matrix $W$ and a latent variable matrix $H$ such that $V \approx WH$. In contrast to other decomposition-based unsupervised learning methods, all three matrices $V, W$ and $H$ are restricted to non-negative entries. While NMF can model data that is inherently non-negative, such has semantic features of images and text, effectively Yuan & Oja (2005); Shahnaz et al. (2006), the method is unable to learn subtractive and non-linear structures that are present in the data Lee & Seung (1999).

Semi-non-negative matrix factorization Ding et al. (2008) relaxes the original restriction to non-negative observations of NMF, by only constraining the weight matrix $W$ to be non-negative. Deep semi-NMF Trigeorgis et al. (2014) further enhances the expressiveness of NMF by adding multiple layers and non-linearities between them to the decomposition.

Concerning this work, the semantics of mono-nets from the second layer on is equivalent to that of deep semi-NMF models. However, the unconstrained first layer of mono-nets provides universal approximation capabilities, enabling mono-nets to learn subtractive and non-monotonic input dependencies. Moreover, while deep NMF models are mostly trained via layer-wise learning in an unsupervised context Trigeorgis et al. (2014); Yu et al. (2018), the sole purpose of mono-nets is to investigate alternatives to backpropagation for training multi-layer classifiers.

## 4 EXPERIMENTS

So far, we have only proven the learning capabilities of mDFA. What remains unclear is whether mDFA can deliver its theoretical promises in practice. In this section, we experimentally evaluate the learning performance of mDFA on a set of empirical benchmarks. We aim to answer the following two questions:

- How well does mDFA perform compared to DFA, FA, and backpropagation in "natural conditions," i.e., in binary classification tasks, and

- how much does the performance of mDFA degrade in multi-class classification tasks?

Our performance measure is the achieved classification accuracy of a network trained by a particular method. First, we report the highest achieved accuracy on the training set, which tells us how well the algorithm could fit the model to the training data. Secondly, for each method, we tuned the hyperparameters on a separate validation set and selected the best performing configuration to be evaluated on the test data. The obtained test accuracy tells us how well the model generalizes to data outside the training set.

We evaluate fully-connected networks (FC) and Convolutional networks (CNNs) in form of modified *all-convolutional-nets* (Springenberg et al., 2015) with tangent hyperbolic, ReLU (Nair & Hinton, 2010), and the hard-tanh non-linearity. The hard-tanh function is defined as

$$\text{hard-tanh}(x) := \min(\max(x, -1), 1). \tag{15}$$

**Hyperparameters** For all training methods, we fixed the batch size to 64, applied no regularization, no normalization, and no data augmentation. Optimizer, i.e., {"Vanilla" Gradient Descent, Adam (Kingma & Ba, 2014), Rmsprop (Tieleman & Hinton, 2012) }, learning rate, training epochs, and weight initialization method were tuned on the validation set. We tested three different weight initialization schemes; all zeros, a scaled uniform, and a normal distribution. Note that *all zeros* was only tested on the forward weights. Our uniform initialization followed the methodology of Nøkland (2016), i.e., scaling the bounds of the distribution inversely by the square-root of the number of incoming connections of a neuron. In order to comply with the weight constraints of mono-nets, for mDFA the lower bound of the uniform distribution was set to $\varepsilon = 10^{-3}$. Moreover, for mDFA we post-processed the initial weights of the normal distribution by taking their absolute values. Input variables are scaled to [0,1]. Detailed network architectures and a brief discussion about the best performing hyperparameter configurations are listed in the supplementary materials in section B.1 and section B.3.

## 4.1 BINARY CLASSIFICATION

We created a series of binary classification benchmarks by randomly sampling two classes from the well-studied CIFAR-100 and ImageNet datasets. We then train and test a 1-vs-1 classifier on the samples of the two classes. For each dataset, we create five such "binarized" sub-datasets and report the mean and standard deviation over these experiments. CIFAR-100 (Krizhevsky et al., 2014) poses a challenging classification dataset, consisting of 32-by-32 RGB images in 100 real-world object categories. ImageNet (Russakovsky et al., 2015) is a large-scale object classification benchmark and de-facto standard to asses new deep learning methods. Each sample is a high-resolution image representing one out of 1000 possible object classes. We pre-processed all samples by cropping and resizing each image to 224-by-224 pixels. Because ImageNet lacks a public test set, we will report the validation accuracy, i.e., as it was reported by Bartunov et al. (2018).

The results on the binarized benchmarks are shown in Table 1 for CIFAR-100 and Table 2 for ImageNet. mDFA could bring the training error to zero for fully-connected networks, and match the test/validation accuracy of backpropagation for convolutional networks.

**Poor learning performance for ReLU networks** One surprising characteristic in our results is that mDFA fails to provide the same level of learning performance as backpropagation for ReLU networks. Recall that Theorem 1 proves the universal approximation capabilities only for mono-net with tanh activation function. A mono-net with ReLU non-linearity restricts both, the activation and the weight values, to positive values. These constraints arguably limit the approximation capabilities of mono-net in combination with ReLU and thus explain the poor performance of mDFA for ReLU networks. We could validate our hypothesis by testing a rectifier non-linearity that also contains negative values in its image. Coincidentally, the hard-tanh function matches exactly this criterion. Therefore, the decent learning performance of mDFA for hard-tanh networks confirms our hypothesis. Note that also FA and DFA expressed an improvement in performance when switching from ReLU to hard-tanh activation. This observation suggests that feedback alignment algorithms in general benefit from symmetric activation functions.

**Discrepancy between fully-connected and convolutional networks** Though mDFA achieved the same training accuracy as backpropagation for fully-connected networks, the generalization ability, i.e., test and validation accuracy, slightly lacks behind BP in our binarized-CIFAR-100 benchmark. This observation aligns with the studies by Nøkland (2016); Bartunov et al. (2018). For convolutional neural networks, this effect is reversed, i.e., a decent test/validation accuracy but a higher training error than BP. We speculate that the two tested initialization schemes for the feedback weights caused this discrepancy. Both backpropagation and feedback alignment have been shown to be sensitive to the employed initialization method (Zhang et al., 2019; Bartunov et al., 2018). The restriction to positive values of the weights used in mDFA, requires a re-thinking of the initialization methods examined in the literature. However, we will leave this study open for future work.

| Activation | Model | Training accuracy (FC) | Test accuracy (FC) | Training accuracy (CNN) | Test accuracy (CNN) |
|---|---|---|---|---|---|
| hard-tanh | BP | **100.0** $\pm$ 0.0% | **81.3** $\pm$ 11.4% | **100.0** $\pm$ 0.0% | **80.9** $\pm$ 10.4% |
| | DFA | 84.1 $\pm$ 8.2% | 76.8 $\pm$ 11.5% | 84.4 $\pm$ 7.9% | 80.0 $\pm$ 10.3% |
| | FA | 85.3 $\pm$ 8.0% | 76.9 $\pm$ 12.0% | 82.5 $\pm$ 10.2% | 77.5 $\pm$ 12.2% |
| | mDFA | **100.0** $\pm$ 0.0% | 77.3 $\pm$ 12.2% | 87.4 $\pm$ 7.0% | **80.9** $\pm$ 9.2% |
| ReLU | BP | **100.0** $\pm$ 0.0% | **80.0** $\pm$ 10.8% | **100.0** $\pm$ 0.0% | **82.2** $\pm$ 9.5% |
| | DFA | 76.2 $\pm$ 9.6% | 72.9 $\pm$ 9.2% | 73.0 $\pm$ 15.5% | 72.0 $\pm$ 16.2% |
| | FA | 77.9 $\pm$ 10.2% | 71.0 $\pm$ 12.0% | 68.6 $\pm$ 15.9% | 69.8 $\pm$ 15.7% |
| | mDFA | 78.2 $\pm$ 9.9% | 62.1 $\pm$ 12.7% | 78.0 $\pm$ 11.7% | 75.5 $\pm$ 11.7% |
| tanh | BP | **100.0** $\pm$ 0.0% | **81.4** $\pm$ 10.5% | **100.0** $\pm$ 0.0% | **81.1** $\pm$ 11.9% |
| | DFA | 84.2 $\pm$ 8.2% | 77.5 $\pm$ 11.2% | 81.6 $\pm$ 9.3% | 79.7 $\pm$ 10.5% |
| | FA | 84.9 $\pm$ 8.0% | 77.6 $\pm$ 12.2% | 94.2 $\pm$ 3.5% | **83.4** $\pm$ 8.7% |
| | mDFA | **100.0** $\pm$ 0.0% | 77.2 $\pm$ 12.2% | 89.0 $\pm$ 6.0% | **81.1** $\pm$ 9.8% |

Table 1: Accuracies on binarized-CIFAR-100. Mean and standard deviation

| Activation | Model | Training acc. (FC) | Validation acc. (FC) | Training acc. (CNN) | Validation acc. (CNN) |
|---|---|---|---|---|---|
| hard-tanh | BP | **98.4** $\pm$ 3.6% | 77.3 $\pm$ 6.8% | **99.8** $\pm$ 0.5% | 81.7 $\pm$ 11.6% |
| | DFA | 93.8 $\pm$ 13.0% | 74.0 $\pm$ 8.5% | **100.0** $\pm$ 0.0% | 83.0 $\pm$ 8.6% |
| | FA | 97.5 $\pm$ 0.8% | 74.7 $\pm$ 7.7% | 51.9 $\pm$ 0.5% | 54.7 $\pm$ 4.6% |
| | mDFA | 95.3 $\pm$ 10.4% | **79.5** $\pm$ 9.8% | 79.5 $\pm$ 9.8% | **83.8** $\pm$ 7.2% |
| ReLU | BP | **100.0** $\pm$ 0.0% | **78.3** $\pm$ 7.7% | **90.0** $\pm$ 18.1% | **72.5** $\pm$ 13.7% |
| | DFA | 75.1 $\pm$ 8.1% | 73.2 $\pm$ 8.0% | 50.4 $\pm$ 0.9% | 52.7 $\pm$ 2.4% |
| | FA | 68.3 $\pm$ 7.9% | 71.0 $\pm$ 6.7% | 52.3 $\pm$ 0.9% | 54.8 $\pm$ 2.1% |
| | mDFA | 56.9 $\pm$ 0.9% | 59.7 $\pm$ 2.4% | 52.6 $\pm$ 1.6% | 58.4 $\pm$ 4.5% |
| tanh | BP | **99.9** $\pm$ 0.2% | 75.8 $\pm$ 7.7% | **99.9** $\pm$ 0.1% | 76.2 $\pm$ 9.8% |
| | DFA | 97.9 $\pm$ 2.2% | 76.8 $\pm$ 7.2% | 99.3 $\pm$ 1.5% | **83.2** $\pm$ 8.8% |
| | FA | 98.1 $\pm$ 2.3% | 76.5 $\pm$ 7.2% | 62.4 $\pm$ 17.3% | 60.8 $\pm$ 13.8% |
| | mDFA | **99.9** $\pm$ 0.1% | **78.0** $\pm$ 6.8% | 78.2 $\pm$ 9.4% | **83.0** $\pm$ 6.8% |

Table 2: Accuracies on binarized-ImageNet. Mean and standard deviation

| Set | Classes | BP | DFA | FA | mDFA |
|---|---|---|---|---|---|
| Training | 3 | **100.0** $\pm$ 0.0% | 50.5 $\pm$ 9.5% | 79.0 $\pm$ 6.1% | 53.3 $\pm$ 8.9% |
| | 5 | **100.0** $\pm$ 0.0% | 33.8 $\pm$ 9.7% | 68.4 $\pm$ 6.0% | 34.1 $\pm$ 9.8% |
| | 10 | **100.0** $\pm$ 0.0% | 24.7 $\pm$ 1.8% | 52.4 $\pm$ 4.0% | 25.9 $\pm$ 1.2% |
| | 25 | **99.9** $\pm$ 0.1% | 13.4 $\pm$ 1.7% | 31.3 $\pm$ 1.6% | 13.1 $\pm$ 1.4% |
| | 50 | **99.7** $\pm$ 0.1% | 8.5 $\pm$ 0.1% | 20.0 $\pm$ 1.1% | 7.9 $\pm$ 0.2% |
| Test | 3 | **77.4** $\pm$ 8.7% | 49.1 $\pm$ 7.2% | **74.2** $\pm$ 8.1% | 50.9 $\pm$ 7.2% |
| | 5 | **68.5** $\pm$ 8.1% | 31.2 $\pm$ 11.6% | **60.7** $\pm$ 8.4% | 33.9 $\pm$ 9.8% |
| | 10 | **57.3** $\pm$ 4.9% | 25.0 $\pm$ 3.2% | 46.9 $\pm$ 3.8% | 26.3 $\pm$ 2.3% |
| | 25 | **41.2** $\pm$ 2.6% | 13.4 $\pm$ 1.9% | 28.2 $\pm$ 1.4% | 13.1 $\pm$ 1.3% |
| | 50 | **31.8** $\pm$ 1.8% | 8.8 $\pm$ 0.4% | 18.1 $\pm$ 0.9% | 8.0 $\pm$ 0.4% |

Table 3: Mulit-class classification accuracies of fully-connected network with tanh activation on $n$-class CIFAR-100. Mean and standard deviation.

## 4.2 MULTI-CLASS CLASSIFICATION

Here we modify the classification benchmark creation procedure used above, by randomly sampling $n$ classes from the datasets instead of just two. Due to their compelling results in our binary classification benchmark, we restrict our evaluation to networks with tanh activation. Table 3 and Table 4 show the results on our $n$-class CIFAR-100 benchmark for fully-connected and CNNs respectively. The results on $n$-class ImageNet can be found in the supplementary materials in Table 5 and 6.

**mDFA can provide learning for networks with more than one output neuron** Though we do not have a complete theory on mDFA for multi-dimensional outputs, our experiments indicate that mDFA can provide learning for networks with more than one output neuron. In particular, the achieved accuracies of mDFA outperforms other feedback alignment methods for convolutional networks with ten or fewer output neurons. However, mDFA falls behind standard DFA for networks with more than ten output neurons. This observation suggests that our restriction to positive weights can be beneficial even for multi-class tasks but eventually hurts learning performance when the number of classes grows larger.

**Feedback alignment algorithms, in general, tend to struggle with increasing output dimension** Our results express the trend that with an increase in the number of classes, feedback alignment algorithms struggle to fit the training data. While BP can reduce the training error to almost zero independently of the output dimension, the training errors achieved by FA algorithms are significantly higher and correlate with the number of output neurons. Our observations suggest that the dimension of the error signal affects the training convergence for FA algorithms, in contrast to BP, which appears to be less affected by the dimension of the error signal. This relation potentially provides a step towards understanding why FA algorithms fail on challenging datasets as described by Bartunov et al. (2018).

| Set | Classes | BP | DFA | FA | mDFA |
|---|---|---|---|---|---|
| | 3 | **100.0** ± 0.0% | 86.3 ± 5.0% | 84.1 ± 10.2% | **100.0** ± 0.0% |
| | 5 | **100.0** ± 0.0% | 84.6 ± 4.6% | 74.3 ± 8.6% | **100.0** ± 0.0% |
| Training | 10 | **100.0** ± 0.0% | 83.0 ± 3.5% | 57.9 ± 3.4% | 94.4 ± 1.7% |
| | 25 | **99.9** ± 0.0% | 81.0 ± 1.5% | 36.3 ± 1.8% | 64.8 ± 1.4% |
| | 50 | **98.9** ± 0.0% | 80.5 ± 1.0% | 22.7 ± 1.4% | 44.5 ± 0.9% |
| | 3 | 76.2 ± 10.1% | 79.8 ± 6.2% | 79.6 ± 9.7% | **82.0** ± 7.6% |
| | 5 | 65.4 ± 9.8% | 71.8 ± 7.6% | 70.1 ± 9.0% | **73.3** ± 8.8% |
| Test | 10 | 57.5 ± 5.2% | **65.0** ± 4.8% | 55.1 ± 4.1% | 63.9 ± 4.2% |
| | 25 | 41.8 ± 2.2% | **48.3** ± 2.2% | 35.0 ± 1.4% | 44.0 ± 2.5% |
| | 50 | 33.3 ± 1.5% | **37.1** ± 1.5% | 22.3 ± 1.1% | 34.5 ± 1.5% |

Table 4: Mulit-class classification accuracies of Convolutional Neural Network with tanh activation on $n$-class CIFAR-100. Mean and standard deviation.

## 5 CONCLUSION

Feedback alignment algorithms are promising biologically motivated alternatives to backpropagation. While existing literature provides empirical evidence that FA algorithms can work well in practice, there is still a lack of rigorous theory formalizing their learning capabilities. Here we contributed to the field of biologically motivated learning algorithms, by introducing the first feedback alignment algorithm that provable provides learning for non-linear networks of arbitrary depth and single output neuron. We showed that our FA algorithm outperforms existing FA algorithms in binary classification tasks, and even provide decent learning on multi-class problems.

**Limitations** We demonstrated on empirical benchmarks as well as theoretical examples that our method is limited to networks with scalar output. Indeed, uncovering the mathematical principles behind the decent learning performance of FA and DFA on multi-class tasks remains an open challenge.

**Is this really useful?** In terms of scientific significance, this work provided theoretical contributions toward understanding the capabilities and limits of feedback alignment algorithms. From a practical point of view, our mDFA algorithm has an advantage over backpropagation concerning training latency, as all weight updates can be computed in parallel, i.e., see Figure 1. Furthermore, we have shown that mDFA is superior to other feedback alignment algorithms in binary classification tasks. Consequently, mDFA provides an effective solution for binary classification problems with training latency constraints. *Dynamic branch prediction* in microprocessor pipelines poses such problem instance, where program specific binary branch outcomes, i.e., branch taken/not taken, need to be learned in real-time. Because of this real-time constraint, existing branch predictors often employ only shallow perceptron modules (Jiménez & Lin, 2002; Egan et al., 2003). mDFA could enable deeper branch predictor networks to be learned in real-time.

### ACKNOWLEDGMENTS

This research was supported in part by the Austrian Science Fund (FWF) under grant Z211-N23 (Wittgenstein Award).

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

## A APPENDIX

### A.1 MONO-NETS ARE UNIVERSAL APPROXIMATORS

We denote

$$\sigma(x) := \frac{1}{1 + \exp(-x)}$$

the *sigmoid* function,

$$\tanh(x) := \frac{e^x - e^{-x}}{e^x + e^{-x}}$$

the *hyperbolic tangent* and

$$\mathbb{1}[expr] := \begin{cases} 1 & \text{if } expr \text{ is true} \\ 0 & \text{if } expr \text{ is false} \end{cases}$$

the *indicator function*.

From the definitions above we can derive the equalities

$$\sigma(x) = \frac{1}{2}\left(1 + \tanh(\frac{x}{2})\right) \tag{16}$$

$$\tanh(x) = -\tanh(-x) \tag{17}$$

, which we will need for our proof.

**Lemma 1** (Universal Approximation Theorem). *For any given $f \in C(I_n)$ and $\varepsilon > 0$, there exist a function $g : I_n \to \mathbb{R}$ of the form*

$$g(x) := \sum_{i=1}^{N} v_i \sigma(w_i^T x + b_i) \tag{18}$$

*with $v \in \mathbb{R}^N, w_i \in \mathbb{R}^n, b \in \mathbb{R}^N$ and $N < \infty$ such that*

$$\|g(x) - f(x)\|_\infty < \varepsilon.$$

*In essence, the set of functions $g(x)$ of the form given in (18) is* dense *in $C(I_n)$.*

*Proof.* See Hornik et al. (1989) $\qquad \square$

**Theorem 3** (mono-nets are Universal Approximators). *For any given $f \in C(I_n)$ and $\varepsilon > 0$, there exist a function $m : I_n \to \mathbb{R}$ of the form*

$$m(x) := \sum_{i=1}^{M} v_i \tanh(w_i^T x + b_i) + c \tag{19}$$

*with $v \in \mathbb{R}_+{}^M, w_i \in \mathbb{R}^n, \bar{b} \in \mathbb{R}^M, c \in \mathbb{R}$ and $M < \infty$ such that*

$$\|m(x) - f(x)\|_\infty < \varepsilon.$$

*In essence, the set of functions $m(x)$ of the form given in (19) is* dense *in $C(I_n)$.*

*Proof.* By Lemma 1 we know that there exist a $g(x)$ with $\|g(x) - f(x)\|_\infty < \varepsilon$ that has the form

$$g(x) = \sum_{i=1}^{N} v_i \sigma(w_i^T x + b_i)$$

We will show that we can reformulate $g(x)$ to the form in equation (19). Our basic idea is to propagate all negative weight entries into to the first layer where negative weight values are allowed.

$$g(x) = \sum_{i=1}^{N} v_i \sigma(w_i^T x + b_i)$$

$$= \sum_{i=1}^{N} v_i \frac{1}{2} \Big( 1 + \tanh(w_i^T x + b_i) \Big)$$

$$= \sum_{i=1}^{N} v_i \frac{1}{2} \tanh \Big( \sum_{j=1}^{n} w_{i,j} x_j + b_i \Big) + \frac{1}{2} \sum_{i=1}^{N} v_i$$

$$= \sum_{i=1}^{N} \mathbb{1}[v_i \geq 0] v_i \frac{1}{2} \tanh \Big( \sum_{j=1}^{n} w_{i,j} x_j + b_i \Big)$$

$$+ \sum_{i=1}^{N} \mathbb{1}[v_i < 0] v_i \frac{1}{2} \tanh \Big( \sum_{j=1}^{n} w_{i,j} x_j + b_i \Big)$$

$$+ \frac{1}{2} \sum_{i=1}^{N} v_i$$

$$= \sum_{i=1}^{N} \mathbb{1}[v_i \geq 0] v_i \frac{1}{2} \tanh \Big( \sum_{j=1}^{n} w_{i,j} x_j + b_i \Big)$$

$$+ \sum_{i=1}^{N} -\mathbb{1}[v_i < 0] v_i \frac{1}{2} \tanh \Big( - \sum_{j=1}^{n} w_{i,j} x_j - b_i \Big)$$

$$+ \frac{1}{2} \sum_{i=1}^{N} v_i$$

$$= \sum_{i=1}^{N} \bar{v}_i \tanh \Big( w_i^T x + b_i \Big)$$

$$+ \sum_{i=1}^{N} \bar{v}'_i \tanh \Big( \bar{w}_i^T x + \bar{b}_i \Big) + c$$

with

$$\bar{v}_i = \mathbb{1}[v_i \geq 0] v_i \frac{1}{2} \geq 0$$

$$\bar{v}'_i = -\mathbb{1}[v_i < 0] v_i \frac{1}{2} \geq 0$$

$$\bar{w}_i = -w_i^T$$

$$\bar{b}_i = -b_i$$

$$c = \frac{1}{2} \sum_{i=1}^{N} v_i$$

Therefore, we showed that there exist a $m(x)$ with a form as in equation (19) that satisfy

$$\|m(x) - f(x)\|_\infty < \varepsilon.$$

$\square$

This proof showed us how to translate any neural network with sigmoid activation function and one hidden layer of size $N$ into a mono-net with $2N$ hidden units.

A.2   FOR 1-DIMENSIONAL OUTPUTS MDFA COMPUTES THE SIGN OF THE GRADIENT

**Lemma 2** (The gradient of a monotone layer is non-negative). *Let* $m : \mathbb{R}^{n_0} \to \mathbb{R}$ *with* $m(x) :=$ $h_L \circ h_{L-1} \circ \cdots \circ h_2 \circ h_1 \circ h_0(x)$ *be a mono-net according to Definition 1. Then*

$$\Big( \frac{dh_l}{dh_{l-1}} \Big)_{i,j} \geq 0, \tag{20}$$

*for any* $l, i, j$ *with* $2 \leq l \leq L$ *and* $1 \leq i \leq n_l$ *and* $1 \leq j \leq n_{l-1}$

*Proof.* We have to distinguish two cases:
**Case 1:** $l = L$, i.e. there is no activation function.
We have

$$\Big( \frac{dh_l}{dh_{l-1}} \Big)_{i,j} = \Big( W_l \Big)_{i,j} \geq 0, \tag{21}$$

according to the definition in Equation (10).
**Case 2:** $l < L$, i.e. there is an activation function.
We have

$$\Big( \frac{dh_l}{dh_{l-1}} \Big)_{i,j} = \Big( W_l \mathrm{diag}(f'(W_l h_{l-1} + b_l)) \Big)_{i,j} \tag{22}$$

$$= \sum_{k=1}^{n_l} \Big( W_l \Big)_{i,k} \Big( \mathrm{diag}(f'(W_l h_{l-1} + b_l)) \Big)_{k,j}, \tag{23}$$

where $f'$ is the derivative of the activation function. Because $f$ is a monotonic function, its derivative is non-negative everywhere. As a result we have a sum of a product of non-negative values. Ergo

$$\Big( \frac{dh_l}{dh_{l-1}} \Big)_{i,j} \geq 0 \tag{24}$$

□

**Lemma 3** (The gradient of a composition of monotone layers is non-negative). *Let* $m : \mathbb{R}^{n_0} \to \mathbb{R}$ *with* $m(x) := h_L \circ h_{L-1} \circ \cdots \circ h_2 \circ h_1 \circ h_0(x)$ *be a mono-net according to Definition 1. Then*

$$\Big( \frac{dh_l}{dh_k} \Big)_{i,j} \geq 0, \tag{25}$$

*for any* $l, k, i, j$ *with* $2 \leq l \leq L$ *and* $1 \leq k < l$ *and* $1 \leq i \leq n_l$ *and* $1 \leq j \leq n_k$

*Proof.* By applying the chain rule we get

$$\Big( \frac{dh_l}{dh_k} \Big)_{i,j} = \Big( \prod_{m=k+1}^{l} \frac{dh_m}{dh_{m-1}} \Big)_{i,j} \tag{26}$$

According to Lemma 2 we have a product of non-negative values. Because a product of non-negative values is non-negative itself, we have

$$\Big( \frac{dh_l}{dh_k} \Big)_{i,j} \geq 0 \tag{27}$$

□

**Theorem 4** (For 1-dimensional outputs mDFA computes the sign of the gradient). *Let* $\mathcal{L}(y, h_L)$ *be a loss function,* $m(x) := h_L \circ h_{L-1} \circ \cdots \circ h_2 \circ h_1 \circ h_0(x)$ *be a mono-net according to Definition 1 with parameters* $\Theta := \{ W_l, b_l | l = 1, \ldots L \}$. *We denote* $\delta\theta$ *the update value computed by mDFA and* $\nabla\theta$ *as the gradient* $\frac{\partial \mathcal{L}}{\partial \theta}$ *for any* $\theta \in \{W_l, b_l\}$ *with* $1 \leq l \leq L$. *If* $n_L = 1$, *it follows that*

$$\Big( \delta\theta \Big)_{i,j} \cdot \Big( \nabla\theta \Big)_{i,j} \geq 0, \tag{28}$$

*for each coordinate* $(i, j)$ *of* $\theta$.

*Proof.* We distinguish two cases:

**Case 1:** $l = L$, i.e. $\theta$ is a parameter of the last layer. From the definition of mDFA we have

$$\delta\theta := \frac{d\mathcal{L}}{dh_L}\frac{dh_L}{d\theta}. \tag{29}$$

For the gradient by applying the chain rule we get

$$\nabla\theta = \frac{d\mathcal{L}}{d\theta} = \frac{dL}{dh_L}\frac{dh_L}{d\theta} \tag{30}$$

Thus, in the last layer the mDFA update equals the gradient.

**Case 2:** $l < L$, i.e $\theta$ is a parameter of a hidden layer. From the definition of mDFA we have

$$\left(\delta\theta\right)_{i,j} := \left(\frac{d\mathcal{L}}{dh_L}B\frac{dh_l}{d\theta}\right)_{i,j} \tag{31}$$

with $B \in \mathbb{R}_+^{n_L \times n_l}$. Next, we expand the multiplication,

$$\left(\delta\theta\right)_{i,j} = \sum_{k=1}^{n_L}(\frac{d\mathcal{L}}{dh_L})_k B_{k,i}\frac{d(h_l)_i}{d\theta_{i,j}} \tag{32}$$

$$= \frac{d(h_l)_i}{d\theta_{i,j}}\sum_{k=1}^{n_L}(\frac{d\mathcal{L}}{dh_L})_k B_{k,i}. \tag{33}$$

We assumed that the output dimension is 1, i.e. $n_L = 1$. Therefore,

$$\left(\delta\theta\right)_{i,j} = \frac{d(h_l)_i}{d\theta_{i,j}}(\frac{d\mathcal{L}}{dh_L})_1 B_{1,i}. \tag{34}$$

For the gradient by applying the chain rule we get

$$\nabla\theta = \frac{d\mathcal{L}}{d\theta} = \frac{dL}{dh_L}\frac{dh_L}{d\theta} \tag{35}$$

$$= \frac{d\mathcal{L}}{dh_L}\frac{dh_L}{dh_l}\frac{dh_l}{d\theta} \tag{36}$$

Like above, we expand the multiplication,

$$\left(\nabla\theta\right)_{i,j} = \sum_{k=1}^{n_L}(\frac{d\mathcal{L}}{dh_L})_k(\frac{dh_L}{dh_l})_{k,i}\frac{d(h_l)_i}{d\theta_{i,j}} \tag{37}$$

$$= \frac{d(h_l)_i}{d\theta_{i,j}}\sum_{k=1}^{n_L}(\frac{d\mathcal{L}}{dh_L})_k(\frac{dh_L}{dh_l})_{k,i}. \tag{38}$$

We assumed that the output dimension is 1, i.e. $n_L = 1$. Therefore,

$$\left(\nabla\theta\right)_{i,j} = \frac{d(h_l)_i}{d\theta_{i,j}}(\frac{d\mathcal{L}}{dh_L})_1(\frac{dh_L}{dh_l})_{1,i}. \tag{39}$$

For Equation (28) we get by applying Lemma 3,

$$\left(\delta\theta\right)_{i,j}\cdot\left(\nabla\theta\right)_{i,j} = \left(\frac{d(h_l)_i}{d\theta_{i,j}}(\frac{d\mathcal{L}}{dh_L})_1 B_{1,i}\right)\cdot\left(\frac{d(h_l)_i}{d\theta_{i,j}}(\frac{d\mathcal{L}}{dh_L})_1(\frac{dh_L}{dh_l})_{1,i}\right) \tag{40}$$

$$= \underbrace{\left(\frac{d(h_l)_i}{d\theta_{i,j}}(\frac{d\mathcal{L}}{dh_L})_1\right)^2}_{\geq 0}\underbrace{B_{1,i}}_{\geq 0}\underbrace{(\frac{dh_L}{dh_l})_{1,i}}_{\geq 0} \tag{41}$$

$$\geq 0. \tag{42}$$

$\square$

### A.3 For $k \geq 2$-dimensional outputs mDFA updates may not align with the gradient

**Theorem 5** (For $k \geq 2$-dimensional outputs mDFA updates may not align with the gradient)**.** *Let* $\mathcal{L}(y, h_L)$ *be a loss function,* $m(x) := h_L \circ h_{L-1} \circ \cdots \circ h_2 \circ h_1 \circ h_0(x)$ *be a mono-net according to Definition 1 with parameters* $\Theta := \{W_l, b_l | l = 1, \ldots L\}$. *We denote* $\delta\theta$ *the update value computed by mDFA and* $\nabla\theta$ *as the gradient* $\frac{\partial\mathcal{L}}{\partial\theta}$ *for any* $\theta \in \{W_l, b_l\}$ *with* $1 \leq l < L$. *If* $n_L \geq 2$, *there exists the possibility that*

$$\left(\delta\theta\right)_{i,j} \cdot \left(\nabla\theta\right)_{i,j} < 0, \tag{43}$$

*for at least one coordinate* $(i, j)$ *of* $\theta$.

*Proof.* We construct a minimal counterexample consisting of a network with two outputs and one hidden layer of two neurons.

Let

$$W_2 = \begin{pmatrix} 0 & 1 \\ 1 & 0 \end{pmatrix} \tag{44}$$

$$B_2 = \begin{pmatrix} 1 & 0 \\ 0 & 1 \end{pmatrix} \tag{45}$$

$$\frac{\partial\mathcal{L}}{\partial h_L} = \begin{pmatrix} 1 & 0 \\ 0 & -1 \end{pmatrix} \tag{46}$$

$$\frac{\partial h_1}{\partial\theta} = \begin{pmatrix} 1 & 0 \\ 0 & 1 \end{pmatrix} \tag{47}$$

$$\tag{48}$$

Then we have

$$\nabla\theta = \frac{d\mathcal{L}}{dh_L} W_2^T \frac{dh_l}{d\theta} \tag{49}$$

$$= \begin{pmatrix} 1 & 0 \\ 0 & -1 \end{pmatrix} \begin{pmatrix} 0 & 1 \\ 1 & 0 \end{pmatrix} \begin{pmatrix} 0 & 1 \\ 1 & 0 \end{pmatrix} \tag{50}$$

$$= \begin{pmatrix} -1 & 0 \\ 0 & 1 \end{pmatrix} \tag{51}$$

and

$$\delta\theta = \frac{d\mathcal{L}}{dh_L} B_2 \frac{dh_l}{d\theta} \tag{52}$$

$$= \begin{pmatrix} 1 & 0 \\ 0 & -1 \end{pmatrix} \begin{pmatrix} 1 & 0 \\ 0 & 1 \end{pmatrix} \begin{pmatrix} 0 & 1 \\ 1 & 0 \end{pmatrix} \tag{53}$$

$$= \begin{pmatrix} 1 & 0 \\ 0 & -1 \end{pmatrix} \tag{54}$$

$$\tag{55}$$

, which are orthogonal. □

## B Experiment Setup

### B.1 Network architectures

| Dataset | Fully-connected | Convolutional |
|---|---|---|
| CIFAR-100 | 1024,1024 | (96,5,2),(96,3,2),(96,3,1) |
| ImageNet | 1024,1024,1024,1024 | (96,3,2),(96,5,1),(128,3,2), (192,3,1),(192,3,2),(384,3,1) |

Network architectures, layers are separated by commas. Fully-connected column specifies the number of neurons of each layer. Convolutional column specifies the number of filters, kernel size, and stride for each layer.

## B.2 $n$-CLASS IMAGENET

| Set | Classes | BP | DFA | FA | mDFA |
|---|---|---|---|---|---|
| | 3 | **100.0** $\pm$ 0.0% | 59.6 $\pm$ 7.2% | 50.4 $\pm$ 5.9% | 51.4 $\pm$ 4.4% |
| Training | 5 | **100.0** $\pm$ 0.0% | 48.9 $\pm$ 5.0% | 41.4 $\pm$ 5.5% | 41.0 $\pm$ 7.1% |
| | 10 | **99.7** $\pm$ 0.5% | 33.4 $\pm$ 4.5% | 24.7 $\pm$ 3.3% | 38.7 $\pm$ 10.1% |
| | 3 | **74.5** $\pm$ 8.8% | 62.8 $\pm$ 11.9% | 64.2 $\pm$ 10.9% | 62.0 $\pm$ 9.6% |
| Validation | 5 | **63.0** $\pm$ 7.4% | 43.8 $\pm$ 6.2% | 42.3 $\pm$ 7.1% | 37.3 $\pm$ 5.2% |
| | 10 | **41.2** $\pm$ 5.0% | 30.8 $\pm$ 4.9% | 26.0 $\pm$ 4.5% | 33.3 $\pm$ 8.0% |

Table 5: Mulit-class classification accuracy of fully-connected network with tanh activation on $n$-class ImageNet. Mean and standard deviation.

| Set | Classes | BP | DFA | FA | mDFA |
|---|---|---|---|---|---|
| | 3 | **100.0** $\pm$ 0.0% | 100.0 $\pm$ 0.0% | 35.8 $\pm$ 0.7% | 68.1 $\pm$ 8.2% |
| Training | 5 | **100.0** $\pm$ 0.0% | 100.0 $\pm$ 0.0% | 21.4 $\pm$ 0.4% | 60.4 $\pm$ 6.2% |
| | 10 | **100.0** $\pm$ 0.0% | 90.4 $\pm$ 10.0% | 11.3 $\pm$ 0.7% | 42.9 $\pm$ 3.9% |
| | 3 | 68.5 $\pm$ 13.0% | **76.5** $\pm$ 9.9% | 53.3 $\pm$ 1.7% | **74.8** $\pm$ 8.8% |
| Validation | 5 | 61.6 $\pm$ 7.3% | **68.7** $\pm$ 6.3% | 29.0 $\pm$ 2.0% | 61.9 $\pm$ 5.3% |
| | 10 | **52.8** $\pm$ 4.6% | **53.6** $\pm$ 5.3% | 10.6 $\pm$ 0.8% | 44.1 $\pm$ 4.4% |

Table 6: Mulit-class classification accuracy of Convolutional Neural Network with tanh activation on $n$-class ImageNet. Mean and standard deviation.

## B.3 DISCUSSION ON HYPERPARAMETERS

We observed that all FA algorithms yield a more stable convergence with "vanilla" stochastic gradient descent, i.e., no post-processing of the FA updates, than with Adam (Kingma & Ba, 2014) or Rmsprop (Tieleman & Hinton, 2012). This may be non-surprising as these acceleration methods have been developed for gradient-based optimization, whereas FA updates roughly align with the gradients at best.

Furthermore, we observed that FA algorithms achieve a descent validation accuracy after the first few training epochs. However, in contrast to BP, these methods may require over a hundred training epochs to converge fully. This "fast-start-slow-convergence" aligns with the observations made by Lillicrap et al. (2016).

We found that FA, DFA, and mDFA perform best when all forward weights are initialized to zero. Notable exceptions are networks with ReLU activation function which express poor performance with the all-zeros initialization scheme. This poor performance with all-zeros initialization partially explains the poor observed performance of FA and DFA for ReLU networks.

In contrast to Nøkland (2016), we observed that backward weights initialized by a normal distribution perform slightly better than the scaled uniform proposed by Nøkland (2016).

We confirmed the observation made by Bartunov et al. (2018), that feedback alignment algorithms are in general relatively sensitive to hyperparameter choice.

