# OpenReview forum: "Learning representations for binary-classification without backpropagation"
_ICLR.cc/2020/Conference — Accept (Poster)_

### Official Review · AnonReviewer3 · 2019-10-21
**Official Blind Review #3**

**Rating:** 6

**Review:**

------update------

After talking to other reviews and reading the rebuttal, I am convinced that the paper contributes sufficiently to the theoretical understanding of the FA algorithm and should be accepted as a conference paper.

I hope that, in the next revision, the authors could include more about the limitation of their work and potential alternatives to improve the generosity of the proposed method.

-------end of the update------

The paper presented a mono-net which has only positive weights and monotonically increasing functions in between layers except for the first layer, and it can be shown that the proposed mono-net is capable of modelling any continuous functions. The modified Direct Feedback Alignment (DFA) is applied where only signs are used to update the weights. There are several issues and concerns that leads me to the following comments and concerns:

(A) Why is it necessary to construct a monotonic layer which is constrained to only be able to approximate monotonic functions? Please elaborate. If the concern is about the gradient update, then there is better way of constructing such a network with limiting it being only capable of modelling monotonic functions.

(B) If, given the proof in appendix, that the proposed mono-net is indeed a universal function approximator, then why do all the layers on top of the first layer have to contain only positive weights?

Following the proof, given a neural network with only one hidden layer and hyperbolic tanh activation functions, as long as the weights in the layer that is after the tanh functions are all non-negative or non-positive, the neural network is also a universal function approximator.

Since the chosen activation function monotonically increases in the input domain, the sign of the update calculated by DFA is the same as the gradient calculated by the chain rule.

The aforementioned way of constructing a neural network allows it to be able to model to change the monotonicity of the function in between layers when the two sets of weights in consecutive two layers have opposite signs. Also, the update gives the sign of the gradient calculated by backpropagation.

(C) What is the different between the proposed network along with the update rule and a network with only non-negative weights in the layers above the first layer trained with RPROP? They look exactly the same to me.

If so, another perspective of the story is that the paper emposes a non-negative constraint on the weights in a neural network, and this could be used as a baseline.

(D) The proof that gives the universal approximation theorem explicitly defines the squashing function/the activation function to be bounded and this paper follows the setting, which is reflected in the experiments where the proposed network with ReLU activation functions don't work well.

However, the expressiveness of ReLU networks has been shown to be strong and they are also universal approximators. I'd believe that with (B), it might lift the constraints brought by ReLU in the settings proposed by the paper.

(E) I am not sure why for now the number of output units matters that much. The training algorithm can easily ignore other output units when samples of a specific class is presented, then after learning, the algorithm can rank the output units to make predictions.

**Experience Assessment:**

I have read many papers in this area.

**Review Assessment: Checking Correctness Of Derivations And Theory:**

I carefully checked the derivations and theory.

**Review Assessment: Checking Correctness Of Experiments:**

I carefully checked the experiments.

**Review Assessment: Thoroughness In Paper Reading:**

I read the paper thoroughly.

---

> ### Author Response · Authors · 2019-11-08
> **Thank you for your comments, we hope we can address your concerns**
>
> We thank the reviewer for their careful analysis of our manuscript.
> Notably, we appreciate the reviewer's thoughts on the methodological approach and the legibility of the paper.
>
> First of all, we want to clarify the statement made by the reviewer "The modified Direct Feedback Alignment (DFA) is applied where only signs are used to update the weights.", which is not 100% correct.
> The mDFA update does not use the signs. Instead, the signs of the mDFA update equal the signs of the true gradient (at least for networks with a single output neuron), i.e., the angle between the update vector and the gradient stay within 90 degrees of each other.
>
> Secondly, we want to recapitulate that the premise of this paper is to seek alternatives to backpropagation for training multi-layer neural networks. Existing research on that topic consists primarily of empirical studies or requires strong assumptions to give any mathematical learning guarantee.
>
> Finally, we want to respond to the detailed concerns that the reviewer made:
> (A) The monotonic layers are necessary for the DFA update (not to be confused with the "gradient" update). We want the DFA update vector to point in roughly the same direction as the gradient update.
> (B) We want to develop a model that, I: is a universal approximator, and II: can be provably trained without backpropagation.
> We were able to achieve both, I and II, by constraining all layers on top of the first layer to non-negative weight values.
> (C) Our target was particularly that the mDFA update rule and the RPROP update should be the same. The difference is that mDFA does not rely on any weight sharing and only employs a weakened form of a reciprocal error transport. Both are biological implausibilities of the backpropagation-of-error algorithm.
> (D) Indeed, it has been shown that unconstrained networks perform specifically well with ReLU activation [1]. However, we assessed in the paper, that having only non-negative weights (our constraint) and non-negative activations (ReLU), hurts the expressiveness and learning of the network. We agree that resolving this limitation is an interesting problem; nevertheless, it is not the main focus of this paper.
> (E)  Generally, the training performance of a network with backpropagation is not affected by the size of the output layer. However, as we have shown empirically and theoretically, it does matter when training the network with mDFA. For details, see supplementary materials A.3.
>
>
> We hope that we could address most of the reviewer's concerns, and we hope the reviewer reconsiders this paper regarding their recommendation on acceptance.
>
>
> [1] Glorot, Xavier and Bordes, Antoine and Bengio, Yoshua. "Deep sparse rectifier neural networks". AISTAT 2011.

---

### Official Review · AnonReviewer1 · 2019-10-23
**Official Blind Review #1**

**Rating:** 6

**Review:**

This paper presents an approach towards extending the capabilities of feedback alignment algorithms, that in essence replace the error backpropagation weights with random matrices.  The authors propose a particular type of network where all weights are constraint to positive values except the first layers, a monotonically increasing activation function, and where a single output neuron exists (i.e., for binary classification - empirical evidence for more output neurons is presented but not theoretically supported).  This is to enforce that the backpropagation of the (scalar) error signal to affect the magnitude of the error rather than the sign, while preserving universal approximation.  The authors also provide provable learning capabilities, and several experiments that show good performance, while also pointing out limitations in case of using multiple output neurons.

The strong point of the paper and main contribution is in terms of proposing the specific network architecture to facilitate scalar error propagation, as well as the proofs and insights on the topic.  The proposed network affects only magnitude rather than sign, and the authors demonstrate that it can do better than current FA and match BP performance.  This seems inspired from earlier work [1,2] - where e.g., in [2] improvements are observed when feedback weights share the sign but not the magnitude of feedforward nets.

Summarizing, I believe that this research is interesting, and can lead to improvements in FA algorithms that could potentially be more biologically plausible, and offer advantages such as full weight update parallelization (although this is more related to the fixed weights rather than the method per-se given my understanding).  However, this also seems - at the moment - to be of limited applicability.

===
Furthermore, the introduction of the network with positive weights in the 2nd layer and on is remiscent of non-negative matrix factorization algorithms.  Can the authors establish a link to these methods, where variants with backprop have also been proposed?



[1] Xiao W. et al. Biologically-Plausible Learning Algorithms Can Scale to Large Datasets, 2018
[2] Qianli Liao, Joel Z Leibo, and Tomaso Poggio.  How important is weight symmetry in backprop-agation, AAAI 2016

**Experience Assessment:**

I have read many papers in this area.

**Review Assessment: Checking Correctness Of Derivations And Theory:**

I assessed the sensibility of the derivations and theory.

**Review Assessment: Checking Correctness Of Experiments:**

I carefully checked the experiments.

**Review Assessment: Thoroughness In Paper Reading:**

I read the paper at least twice and used my best judgement in assessing the paper.

---

> ### Author Response · Authors · 2019-11-08
> **Thanks for your review, we included your suggestions into the paper**
>
> We thank the reviewer for their thoughtful comments on our paper, especially for providing valuable related works, which we incorporated into our manuscript.
>
> We want to respond to two points that the reviewer made:
> 1. We have added a sub-section (2.3 "Sign-symmetry algorithms"), in which we briefly discuss the papers that the reviewer mentioned [1-3]
> 2. We have added a discussion, "How does mDFA relate to the non-negative matrix factorization?" (top of page 6), which provides a comparison to research on non-negative matrix factorization algorithms.
> We have updated the manuscript accordingly, and hope the reviewer supports the contributions of this work on biologically motivated learning algorithms.
>
>
> [1] Liao, Qianli and Leibo, Joel Z and Poggio, Tomaso. "How important is weight symmetry in backpropagation?". AAAI 2016
> [2] Moskovitz, Theodore H and Litwin-Kumar, Ashok and Abbott, LF. "Feedback alignment in deep convolutional networks". arXiv preprint 2018.
> [3] Xiao, Will and Chen, Honglin and Liao, Qianli and Poggio, Tomaso. "Biologically-plausible learning algorithms can scale to large datasets". ICLR 2019

---

### Official Review · AnonReviewer2 · 2019-10-23
**Official Blind Review #2**

**Rating:** 8

**Review:**

This paper examines the question of learning in neural networks with random, fixed feedback weights, a technique known as “feedback alignment”. Feedback alignment was originally discovered by Lillicrap et al. (2016; Nature Communications, 7, 13276) when they were exploring potential means of solving the “weight transport problem” for neural networks. Essentially, the weight transport problem refers to the fact that the backpropagation-of-error algorithm requires feedback pathways for communicating errors that have synaptic weights that are symmetric to the feedforward pathway, which is biologically questionable. Feedback alignment is one approach to solving the weight transport problem, which as stated above, relies on the use of random, fixed weights for communicating the error backwards. It has been shown that in some cases, feedback alignment converges to weight updates that are reasonably well-aligned to the true gradient. Though initially considered a good potential solution for biologically realistic learning, feedback alignment both has not scaled up to difficult datasets and has no theoretical guarantees that it converges to the true gradient. This paper addresses both these issues.

To address these issues, the authors introduce two restrictions on the networks: (1) They enforce “monotone” networks, meaning that following the first layer, all synaptic weights are positive. This also holds for the feedback weights. (2) They require that the task in question be a binary classification task. The authors demonstrate analytically that with these restrictions, direct feedback alignment (where the errors are communicated directly to each hidden layer by separate feedback weights) is guaranteed to follow the sign of the gradient.  (Importantly, they also show that monotone nets are universal approximators.) Empirically, they back up their analysis by demonstrating that in fully connected networks that obey these two restrictions they can get nearly as good performance as back propagation on training sets, and even better performance on tests sets sometimes. However, they also demonstrate (empirically and analytically) that violating the second requirement (by introducing more classes) leads to divergence from the gradient and major impairments in performance relative to backpropagation.

Ultimately, I think this is a great paper, and I think it should be accepted at ICLR. It provides some of the first rigorous analysis of feedback alignment since the original paper came out, and unlike those original analyses, it is not restricted to linear networks (which are certainly not universal function approximators). I have looked over the proofs, and they seem to all be correct. As well, I found the paper easy to read, which was nice. However, there are a few things that could be done to clarify the contributions and situate the work within the field of biological learning algorithms better:

1) Though they do not include rigorous analyses, two previous papers have demonstrated empirically that feedback alignment works extremely well as long as the feedback weights share the same sign as the feedforward weights (see: Moskovitz, Theodore H., Ashok Litwin-Kumar, and L. F. Abbott. "Feedback alignment in deep convolutional networks." arXiv preprint arXiv:1812.06488 (2018) and Liao, Qianli, Joel Z. Leibo, and Tomaso Poggio. "How important is weight symmetry in backpropagation?." In Thirtieth AAAI Conference on Artificial Intelligence. 2016). Due to the requirement for monotone networks, this work is also providing a guarantee that the sign of feedforward and feedback weights are the same. That does not subtract substantially from the contributions of this paper, as the provision of the analytical guarantees is important. But, it is important for the authors to consider how their work relates to this past work. For example, could their analytical approach work equally well with nothing more than a sign symmetry guarantee? This should at least be discussed.

2) It should be admitted somewhere in the paper that the second requirement on the networks for binary tasks is deeply unbiological. As such, it should be recognized in discussion that this paper provides some important contributions to our understanding of feedback alignment, but does not ultimately move the question of biologically realistic learning forward all that much. Indeed, the discussion at the end about applications notably ignores biology. But, rather than just ignoring it, the biological mismatch should be openly admitted.

3) The results with the test sets are a little strange, at least for the tests with larger numbers of categories. In Bartunov et al. (2016), they reported not only better training set results with backprop, but also better test set results generally, than feedback alignment. Are the authors sure that their results, in say, Table 4, are not indicative of insufficient hyperparameter optimization?

Small notes:

- Lillicrap et al.’s paper was eventually published in Nature Communications (see citation above), and the reference should be changed to reflect this.

- The discussion on the impact of convolutions could be beefed up a little bit. In particular, it could be discussed relative to the results of Moskovitz et al. (above) who show that convnets work fine with nothing but guaranteed sign symmetry.

**Experience Assessment:**

I have published in this field for several years.

**Review Assessment: Checking Correctness Of Derivations And Theory:**

I carefully checked the derivations and theory.

**Review Assessment: Checking Correctness Of Experiments:**

I carefully checked the experiments.

**Review Assessment: Thoroughness In Paper Reading:**

I read the paper thoroughly.

---

> ### Author Response · Authors · 2019-11-08
> **Thanks for your review, we incorporated your feedback.**
>
> We thank the reviewer for their thorough review of our paper, and their strong support on this research topic.
>
> We want to respond to the three suggestions the reviewer made:
> 1) We agree that the cited papers provide background and context to our work. Consequently, we have added the sub-section 2.3 "Sign-symmetry algorithms" to our revised submission, where we briefly discuss the papers [1-3]
>
> 2) Agree. We added a statement about it at the end of the discussion **What about networks with more than one output neuron?** (bottom of page 5).
>
> 3) The general trend observed in our experiments approximately aligns with the results of Bartunov et al. (2018), e.g., the accuracies achieved by the fully-connected network trained with BP on CIFAR-10.
> However, several subtle differences explain the discrepancies between our results and the ones reported by Bartunov et al.:
> - We evaluated 10-class subsampled CIFAR-100, whereas Bartunov et al. used standard CIFAR-10 (explains high standard deviation in our results)
> - We performed a proper training-validation-test split, whereas Bartunov et al. reported the best-achieved test accuracy.
> - The Fully-connected network of Bartunov et al. has three hidden layers (ours has only two)
> - The CNN of Bartunov et al. is larger, i.e., 256 filters in the last layer (ours has only 96) and has an additional fully-connected layer between the last convolutional layer and the output layer (ours is an "all-convolutional-net")
> - As the main contribution of Bartunov et al. was to benchmark various biologically inspired learning algorithms, they performed a more extensive hyperparameter search.
> This difference is further amplified for FA and DFA, because of the observation reported in Bartunov et al. (also confirmed in our experiments) that all FA variants are relatively sensitive to hyperparameter choice.
> In case there remain any concerns, the code to reproduce our results is publicly available.
>
> The paper has been updated according to points 1 and 2.
>
> We want to thank the reviewer for their time, and we hope that we could address the concerns of the reviewer.
>
> Remark: Reference to Lillicrap et al. has been updated.
>
> [1] Liao, Qianli and Leibo, Joel Z and Poggio, Tomaso. "How important is weight symmetry in backpropagation?". AAAI 2016
> [2] Moskovitz, Theodore H and Litwin-Kumar, Ashok and Abbott, LF. "Feedback alignment in deep convolutional networks". arXiv preprint 2018.
> [3] Xiao, Will and Chen, Honglin and Liao, Qianli and Poggio, Tomaso. "Biologically-plausible learning algorithms can scale to large datasets". ICLR 2019

---

### Decision · Program_Chairs · 2019-12-19

**Decision:**

Accept (Poster)

**Comment:**

This paper provides a rigorous analysis of feedback alignment under two restrictions 1) that all, except the first, layers are constrained to realize monotone functions and 2) the task is binary classification. Overall, all reviewers agree that this is an interesting submission providing important results on the topic and as such all agree that it should feature at the ICLR program. Thus, I recommend acceptance. However, I ask the authors to take into account the reviewers' concerns and include a discussion about limitations (and general applicability) of this work.